# Measuring dimensionality and purity of high-dimensional entangled states

Isaac Nape [1✉], Valeria Rodríguez-Fajardo [1], Feng Zhu [2], Hsiao-Chih Huang [3], Jonathan Leach [2] & Andrew Forbes [1]

High-dimensional entangled states are promising candidates for increasing the security and encoding capacity of quantum systems. While it is possible to witness and set bounds for the entanglement, precisely quantifying the dimensionality and purity in a fast and accurate manner remains an open challenge. Here, we report an approach that simultaneously returns the dimensionality and purity of high-dimensional entangled states by simple projective measurements. We show that the outcome of a conditional measurement returns a visibility that scales monotonically with state dimensionality and purity, allowing for quantitative measurements for general photonic quantum systems. We illustrate our method using two separate bases, the orbital angular momentum and pixels bases, and quantify the state dimensionality by a variety of definitions over a wide range of noise levels, highlighting its usefulness in practical situations. Importantly, the number of measurements needed in our approach scale linearly with dimensions, reducing data acquisition time significantly. Our technique provides a simple, fast and direct measurement approach.

[1] School of Physics, University of the Witwatersrand, Wits, South Africa. [2] School of Engineering and Physical Sciences, Heriot-Watt University, Edinburgh, UK. [3] Department of Physics, National Taiwan University, Taipei, Taiwan. ✉email: isaacnape@gmail.com

High-dimensional entangled states are widely used throughout quantum science to increase secure information bandwidth and security bounds for quantum communication[1]. Through the precise control of high-dimensional photonic states[2], i.e., time–energy, transverse momentum, spatial degrees of freedom or all of them simultaneously[3], the potential benefits of high-dimensional state encoding are taking centre stage. Recent developments in this direction have displayed the feasibility of quantum information processing that is robustness against optimal quantum cloning machines[4,5], environmental noise[6] and improved information rates[7], demonstrating a significant advantage in comparison to traditional qubit encoding.

Despite the advantages of high-dimensional quantum states, certifying and quantifying the dimensionality of such systems still remains challenging, particularly in the presence of noise. The intuitive approach of simply measuring the width of the modal spectrum is a necessary, but not sufficient condition to determine dimensionality as it fails to account for non-local correlations. Consequently, many techniques have been developed to witness, bound and attempt to quantify high-dimensional quantum states. These include approximating the density matrix via quantum state tomography (QST) with multiple qubit state projections[8], using mutually unbiased bases[9,10] to probe the states or incorporating self-guided approaches[11,12], and testing non-local bi-photon correlations by generalised Bell tests in higher dimensions[13–15]. However, the spectrum measurements do not confirm entanglement, the QST approach scales unfavourably with dimension, only bounds or witnesses are possible with the mutually unbiased bases method and the dimension to be probed must be known a priori (e.g. valid for prime or prime power dimensions) and, finally, the high-dimensional Bell tests can fail the fair sampling condition[16,17]. A further limitation in the present state of the art is that certain dimensionality measurements consider only pure states[9,18], yet noise mechanisms always introduce some degree of the mixture to the system[19], which has a detrimental effect on the accuracy of measured dimensions due to the reduced purity[20]. Yet, knowing the purity and dimension of the state is crucial for fundamental tests of quantum mechanics as well as for quantum information processing protocols, setting the required violation of inequalities in the former, and the information capacity of the state, the allowed error bounds in secure communication systems and the requirement for entanglement distillation in the latter.

In this work, we present a scheme to simultaneously quantify the dimensionality and purity of a bi-photon high-dimensional entangled state, even in the presence of noise, using the isotropic state as our test example. By measuring coincidence fringes from carefully crafted projective measurements, we are able to accurately measure the dimensionality and purity of our entangled state from the visibility, which is only reproducible by entangled photons. We first outline the concept and theory and then demonstrate it experimentally on states with arbitrary purity and a wide range of dimensions. To show the versatility of our approach, we use it to measure entanglement in the topical photonic orbital angular momentum (OAM) basis, and the pixel (position) basis, commonly used in quantum imaging. With knowledge of the visibilities, purity and dimensionality, we have sufficient information to infer other salient measures. Our quantitative technique is simple, robust and scales favourably (linearly) with dimension, making it ideal for practical implementations of quantum protocols with general high-dimensional photonic quantum entangled states, even under undesired noise conditions.

## Results

**Concept**. The task here is to quantify the effective dimensions and purity of an entangled photonic state. If the state is assumed pure and without noise, then the problem is trivial. Here, we wish to make as few assumptions as possible, and consider the more general case of arbitrary mixed states in the presence of noise. Incorporating noise into the description of high-dimensional states is highly topical of late and very much in its infancy, with a full understanding of its deleterious impact only slowly emerging[6]. In general, the purity of the quantum system, and therefore the entanglement between photon pairs, is reduced due to noise introduced by the source, the environment and/or the detection system, very often in the form of white noise produced by background photons, high dark counts in single-photon detectors and unwanted multiphoton events[20]. We follow convention[6] and model such noisy quantum systems by an isotropic state following:

$$\rho = p|\Psi\rangle\langle\Psi| + \left(\frac{1-p}{d^2}\right)\mathbb{I}_{d^2},\qquad(1)$$

which considers contributions of both the pure, $|\Psi\rangle$, and mixed, $\mathbb{I}_{d^2}$ ($d^2$-dimensional identity operator), parts. Although this will be our target state for extracting the purity and dimensionality, it does not appear in the construction of the analysers nor the measurement procedure itself. As such, the state and the parameters to be extracted may be modified to incorporate other factors, e.g., mode-dependent noise due to the resolution limits of detection devices[21]. The pure part, $|\Psi\rangle = \sum_{i=0}^{d-1}\lambda_j|j\rangle|j\rangle$, can be decomposed using the Schmidt basis states, $|j\rangle|j\rangle \in \mathcal{H}_{d^2}$, with corresponding Schmidt coefficients, $\lambda_j$. A variety of entangled, quantum states (time, energy, position, hybrid and hyperentangled) can be decomposed in this way, thus covering a vast number of cases. Here, $p$ is a parameter that determines the purity of the state, and varies from a maximally mixed (separable) state for $p = 0$ to a completely pure (entangled) state for $p = 1$. The purity of a non-separable $d$-dimensional state is given by $\text{Tr}(\rho^2)$, where $1/d < \text{Tr}(\rho^2) \leq 1$, while the bounds on $p$ are $1/(d+1) < p \leq 1$. Hence, since $1/(d+1) \sim 1/d$ for high-dimensional states, it suffices to use the notion of purity and $p$, interchangeably. We use $K = 1/\sum_j|\lambda_j|^4$ as a measure of the local dimensions of the pure part of the state[22].

Our procedure allows us to quickly establish $K$ and $p$, i.e., the dimensionality of the pure component and its probability. Another common measure of dimensionality is the Schmidt rank[23], which we will denote as $d_{\text{ent}}$. The $d_{\text{ent}}$ refers to the dimensionality of the entire state, not just the pure component, and it is possible to deduce $d_{\text{ent}}$ from our approach through knowledge of $K$ and $p$ (see Supplementary Notes 1 and 6).

As we characterise the pure component of the state and establish the overall purity, the number of required measurements scales linearly with the dimension $d$ of the probed Hilbert space. This provides a significant gain in speed for high-dimensional states.

Thus, our proposed method is a fast, accurate and simple procedure to characterise the properties of two-photon, high-dimensional entangled states.

The working principle of our technique is visualised in Fig. 1a, where a set of custom analysers probe distinct parts of a discrete Hilbert space. We can think of each analyser as a probe that scans a sparse set of modes, reminiscent of a conditional measurement that indicates whether there is entanglement within the subspace or not. By combining the information gathered from a number of such analysers, we infer how many dimensions the state occupies. We will demonstrate this procedure both theoretically and experimentally.

It is instructive to illustrate the concept by example. Consider a pair of photons entangled in their polarisation, energy–time, momentum or in the spatial basis[24], the so-called structured

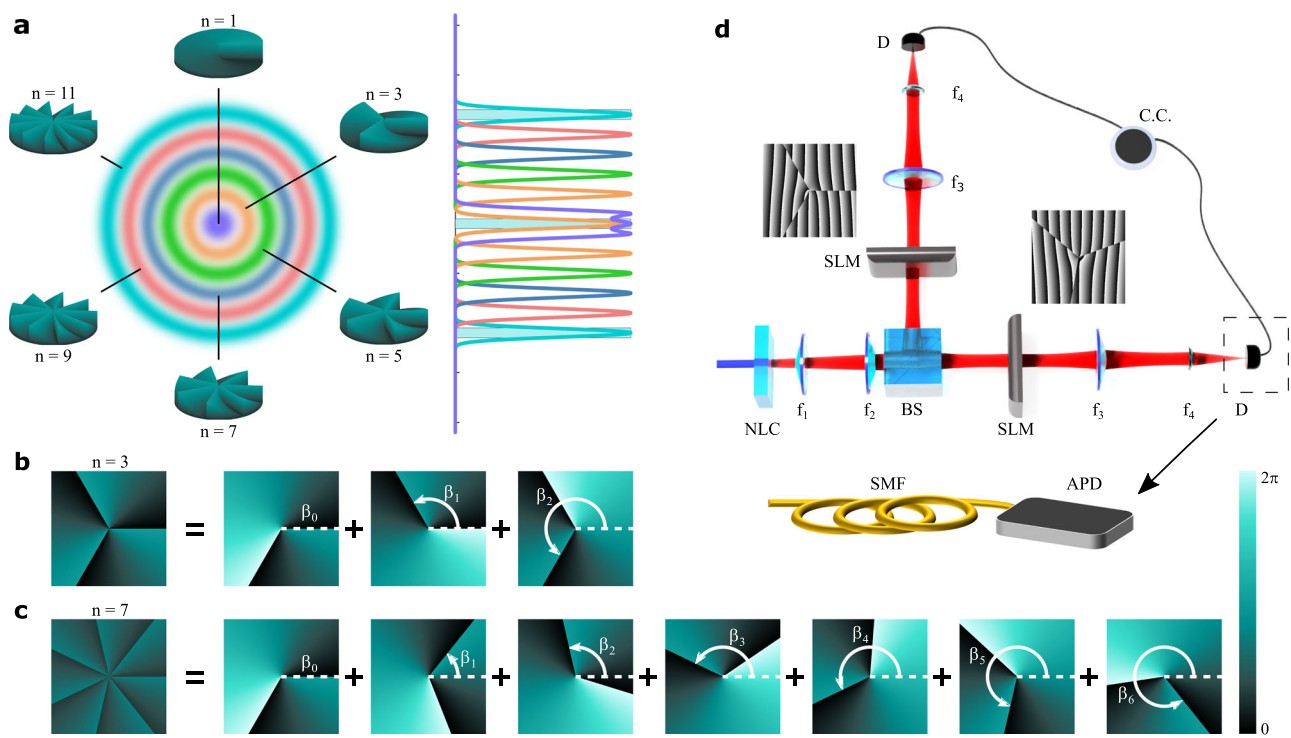

**Fig. 1 Concept and implementation using OAM as an example. a** Conceptual visualisation of different analysers sampling various portions of a high-dimensional discrete Hilbert space. Mode analysers construction for (**b**) $n = 3$ and (**c**) $n = 7$ superpositions of fractional OAM states, where $\beta_i$ is an orientation angle. **d** Schematic of the experimental set-up used to measure the dimensionality and purity of a quantum system. NLC non-linear crystal, $f_{1,2,3,4}$ lens, BS 50:50 beam splitter, SLM spatial light modulator, D detector, APD avalanche photodiode coupled to a single-mode fibre (SMF), CC coincidence counter.

light[25], the former useful to access high dimensions, with up to $100 \times 100$ dimensions already demonstrated[26]. In this work, we will consider two examples: the OAM basis[27,28] and the pixel (position) basis[29]. Due to the great potential of the former, particularly for quantum information processing and communications[30–44], we first illustrate and demonstrate our method for OAM entangled states. In this case, the basis states, $|\ell_1\rangle|\ell_2\rangle$, are associated with an azimuthal phase profile, $\exp(i\ell_{1,2}\phi)$, with $\ell_{1,2} \in \mathbb{Z}$ being the topological charge and $\ell_{1,2}\hbar$ OAM per photon. An OAM entangled pure state can be expressed as

$$|\Psi\rangle = \sum_{\ell_{1,2}=-\infty}^{\infty} \lambda_{\ell_{1,2}} |\ell_1\rangle_A |\ell_2\rangle_B, \qquad (2)$$

where $|\lambda_{\ell_{1,2}}|^2$ is the probability of generating photons in the states $|\pm\ell_{1,2}\rangle$ for photons $A$ and $B$, respectively. For our experimental tests with a Gaussian pumped spontaneous parametric down-conversion (SPDC) source (see "Methods" section), the state only has non-zero probabilities when $\ell \equiv \ell_1 = -\ell_2$. While in general the state, $|\Psi\rangle$, can be represented using an unbounded number of eigenmodes as shown, i.e., $d \rightarrow \infty$, we truncate $|\Psi\rangle$ to $d$ eigenstates. This is simply applying common sense: one should select a Hilbert space with a dimension large enough to test based on what you are looking for (analogous to selecting a camera area that is large enough to fit the image you hope to measure). Importantly, since our approach scales linearly with test dimension, there is no significant penalty for selecting a test dimension that is "too big", in stark contrast to QST-based approaches (see Supplementary Table 2). In this sense, $d$ may be chosen at will.

To gain access to various parts of the Hilbert space, we make use of high-dimensional mode projectors that map onto the states

$$|M, \alpha\rangle_n = \mathcal{N} \sum_{j=0}^{d-1} c_{w_j,M}^n(\alpha) |j\rangle, \qquad (3)$$

where $\mathcal{N}$ is a normalisation factor and $|j\rangle$ are the basis states on the $d$-dimensional space. The coefficients, $c_{w_j,M}^n(\alpha)$, control the amplitudes and phases of the modes in the superposition (see "Methods" section). For OAM basis states, the coefficients can be represented accordingly by replacing the index $w_j$ with the topological charge $\ell = j - (d-1)/2$. Examples of the phase profiles for two such analysers are shown in Fig. 1b, c for $n = 3$ and $n = 7$, respectively, with full details on their construction in the "Methods" section and Supplementary Notes 2–4. While $n$ and $M$ can be chosen arbitrarily, we find it optimal to set $n$ as an odd positive integer and $M = n/2$ (see Supplementary Note 5).

Next, we project each entangled photon onto the superposition states $|M, \theta\rangle_n$ and $|-M, 0\rangle_n$, respectively, where $\theta = [0, \pi/n]$ controls the relative phases between the modes in the superposition. In the context of OAM, this translates into a relative rotation by an angle $\theta$. A typical experimental set-up for implementing this is sketched in Fig. 1d. Entangled photon pairs are generated in a non-linear crystal (NLC) and subsequently projected onto the states $|M, \theta\rangle$ and $|-M, 0\rangle$ by means of holograms programmed onto spatial light modulators (SLMs) having the transmission functions $U_n(\phi; \theta)$ and $U_n^*(\phi; 0)$, respectively. In the OAM degree of freedom, the holograms correspond to fractional OAM modes[45], which are known to have a non-integer azimuthal phase gradient. The modulated photons are then coupled into single-mode fibres and measured in coincidences. The outcome probability of such a measurement,

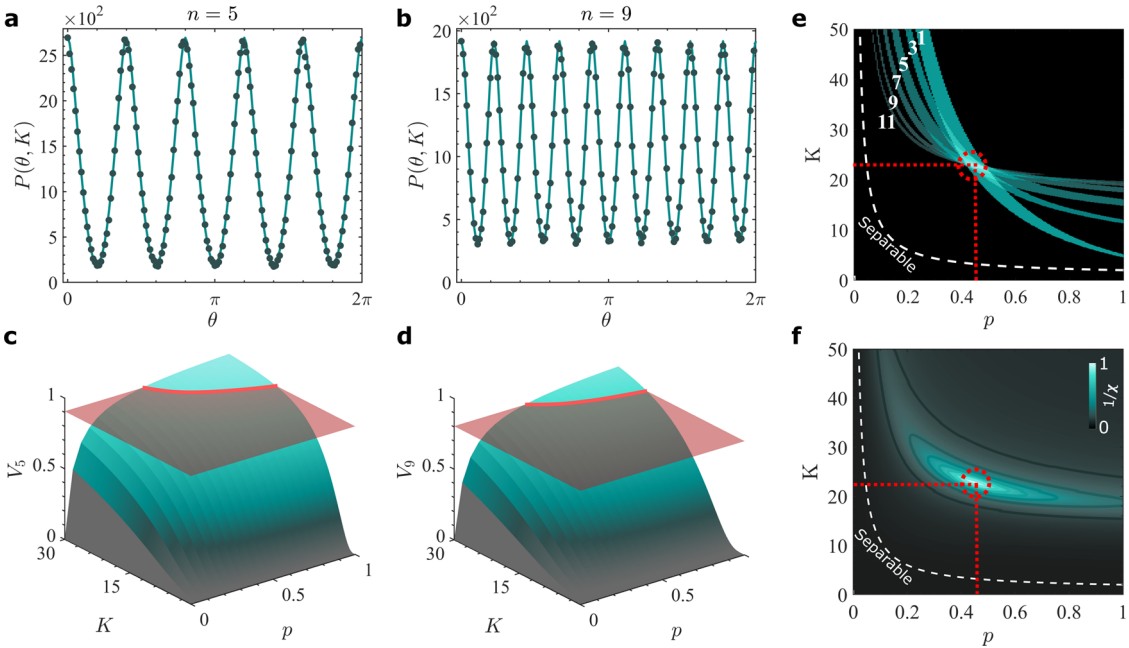

**Fig. 2 Visibility, dimensionality and purity.** Experimental (points) and theoretical (solid lines) coincidence count rates resulting from projections of photons A and B onto the states $|M, \theta\rangle_n$ and $|-M, 0\rangle_n$, respectively, as a function of the relative orientation angle $\theta$ for **a** $n = 5$ and **b** $n = 9$. Theoretical visibility as a function of the dimensionality ($K$) and purity ($p$) for **c** $n = 5$ and **d** $n = 9$, exemplifying it increases monotonically with both parameters. The (red) planes intersecting the curves are the experimental visibilities, with the possible solution space for each shown as a red trajectory. The resulting trajectories for $n = 1, 3, ..., 11$ are shown in (**e**), where the thickness of each is due to the uncertainty in the visibility outcome. The dimension and purity of the system are found where they coincide, shown as a dashed red circle. **f** The latter corresponds to the optimal ($p, K$) that minimise the function $\chi^2(p, K)$, or, equivalently, maximises $\sqrt{1/\chi^2}$, where the minimum of $\chi^2$ is now shown as a peak corresponding to ($p, K$) = (0.45 ± 0.03, 22.84 ± 0.62). The critical bound, $p \leq 1/(K + 1)$, separating entangled and separable states is marked by the white dashed line.

i.e., $|\langle 0, -M|_n \langle \theta, M|_n \rho |M, \theta\rangle_n |-M, 0\rangle_n|$, is

$$P_n(\theta; p, K) = p P_n(\theta, K) + \frac{1 - p}{K^2} I_n(0, K), \quad (4)$$

where $I_n(0, K)/K^2$ is the probability resulting from the overlap of the analysers with the maximally mixed state and $P_n(\theta, K) = \left| \sum_{\ell=-(K-1)/2}^{(K-1)/2} \lambda_\ell c_{\ell,M}^n(0) c_{-\ell,-M}^n(\theta) \right|^2$ is the overlap probability with the pure state, with $M = n/2$ the fractional charge and $\lambda_\ell$ the initial bi-photon OAM spectrum. For a pure state, the probability curves have a parabolic shape following $P_n(\theta) = (\pi(2t - 1) - n\theta)^2/\pi^2$, where $t = 1, 2, ..., n$. In Fig. 2a, b, we show as solid lines the theoretical probabilities (calculated using Eq. (4)) of such a measurement as function of $\theta$.

We choose odd values of $n$ and $M = n/2$ to ensure high visibility, which increases monotonically with $K$ and $p$ for each analyser (see Supplementary Note 6 and Supplementary Fig. 5). In general, both the shape and visibility of the fringes yield information about the state. To make the approach accurate and precise, we measure several visibilities, $V_n$, for $n = 1, 3, 5, ..., 2N - 1$, and infer the state properties by the intersection of their solution spaces (Fig. 2e).

**Orbital angular momentum basis measurements.** The set-up used to demonstrate our scheme is shown conceptually in Fig. 1d with the corresponding detailed description in the "Methods" section. We measure the coincidences between the signal and idler photons for analyser projections on both arms as a function of the relative rotation angle of the holograms. To achieve this, we encoded the fractional OAM mode analyser on the SLM in the signal arm fixed at an angle $\theta = 0$, while the conjugate mode was encoded in the idler arm and rotated at angles $\theta \in [0, 2\pi]$.

To illustrate the operation of our technique, we measured the coincidence rates for six ($N = 6$) analysers with $n = 1, 3, 5, 7, 9$ and $11$, and $M = n/2$, with example outcomes for $n = 5$ and $n = 9$ shown as filled circles in Fig. 2a, b, respectively. No background subtraction was performed on the measurements to leave noise in the system, which was deliberately increased (see "Methods" section for experimental conditions) to enact a range in purities for test purposes. Importantly, the periodicity in the detected probabilities confirms the azimuthal $n$-fold symmetry predicted by our theory (solid curve). Because the visibility is a monotonically increasing function of dimension and purity, measured visibility returns a range of possible ($p, K$) values, a "trajectory" or curve in the ($p, K$) space. This is illustrated in Fig. 2c, d, where the measured visibility (red horizontal plane) intercepts the visibility function along a curve (red curve) that restricts the possible solutions, $K$ and $p$, to those consistent with the measurement outcome. The set of such curves from measuring many visibilities (each with its own analyser/projection) then restricts the final solution to a narrow region in ($p, K$), whose uncertainty (width) is determined primarily from the uncertainly in the visibility measurement. An example is shown in Fig. 2e, where each solution trajectory is projected onto the ($p, K$) plane. Final values and uncertainty of ($p, K$) can be determined by an appropriate routine to find the interception of all such trajectories by a minimisation procedure, as shown in Fig. 2f.

Using this approach, we infer the purity and dimensionality of the system to be ($p, K$) = (0.45 ± 0.03, 22.84 ± 0.62).

In Fig. 3a we show the six measured visibilities as square data points together with the calculated visibility (solid red line) based on the inferred ($p, K$), which clearly match very well. This is confirmation of the minimisation procedure for finding the intercept. In order to assess the procedure under high noise levels,

we introduced background noise using a white light source and repeated the measurements, shown as the circle data points and the associated blue dashed line in Fig. 3. The average quantum contrast (see Supplementary Note 8), measured from the spiral spectrum in Fig. 3c, d, dropped from $Q = 19.19$ to 3.76, resulting in a reduced purity and dimensionality of $(p, K) = (0.13 \pm 0.01, 17.73 \pm 0.71)$. Note that the minimisation was performed over the parameters $(p, K)$, but additional parameters could also be added, e.g., to take account of the modal cross-talk in the observed data. In our case, we choose minimisation over a small set of parameters in order to keep the method and model simple.

As a form of validation of these results, we estimate values from other techniques, with the comparison given in Table 1. If the dimension and noise are known or assumed, then it is possible to calculate the purity following $\hat{p} = (Q - 1)/(Q - 1 + K)$, where $Q$ is the quantum contrast and $K$ the dimension[20]. Likewise, if the state is assumed to be pure and not mixed, and background subtraction is done to remove noise, then the spiral spectrum can be used to get an upper bound on the dimension. For the two noise cases in Table 1, low and high, we find purity estimates of $\hat{p} \approx 0.44 \pm 0.01$ and $\hat{p} \approx$

0.13 ± 0.02 from estimates of the dimensionality of $\hat{K} \approx 22 \pm 1$ and $\hat{K} \approx 18 \pm 1$, respectively. These values are in excellent agreement with our results, which did not require any such assumptions, nor any noise adjustments.

**Pixel basis measurements**. To illustrate that OAM is only an example and that the approach is general, we perform the same procedure using the pixel basis, shown in Fig. 4a. Here, the spatial basis is position as "pixels" in the transverse plane, with the number of pixels setting the test dimension. The size and number would be judiciously chosen based on the source of biphotons and the imaging resolution of the optical system. We use grids from $3 \times 3$ up to $11 \times 11$, thus testing to over 100 dimensions. Holograms for three analyser cases are shown in Fig. 4b for the 81 dimensional example, where the phases within the $9 \times 9$ pixel grid are shown to change. Although there is no resemblance to the prior OAM holograms, the measurement procedure is identical. From the resulting visibilities, we again infer the key parameters from the intersection of the trajectories in $(p, K)$ space, shown visually in Fig. 4c, d.

Our approach has the benefit of a wealth of information in the analyser visibilities, as well as knowledge of both $K$ and $p$. This is sufficient to infer other key information (see Supplementary Fig. 7), such as an estimate of the state fidelity ($F_p$) and the Schmidt rank, $d_{ent} = K \times F_p$ (see Supplementary Note 10 and ref. [23]). We show the outcome for this measurement in Fig. 4e, where the "effective dimensions" as measured by the Schmidt rank decreases as the purity decreases, becoming separable below a critical value that corresponds to the separability boundaries shown in Fig. 2e, f. Our data are shown with deliberately introduced noise to reduce the purity, and juxtaposed with the case of background subtraction to eliminate the noise. Here, we see that the Schmidt rank gives a lower bound for the system. Our technique can detect correlations below the separability criterion for isotropic states, meaning that it is sensitive to correlations even in extreme noise situations, which may prove valuable in situations such as high-resolution quantum imaging in real-world scenarios[46].

## Discussion

A quantitative measure of dimensionality and purity, particularly in the presence of (inevitable) deleterious noise that degrades the purity, is crucial for many quantum protocols and studies. For example, there is a minimum purity needed to witness entanglement in a given dimension[47–49], setting the transition from separable to entangled states. Likewise, knowing the purity is important in entanglement distillation processes since it informs whether the noise can be removed for a given dimension[50,51], while in entanglement-based quantum communication there is a minimum purity[52] associated with security[53]. In turn, the dimensionality sets the information capacity of the state for quantum information processing and the error tolerance in quantum communication protocols, while high-dimensional states are important for fundamental tests of quantum mechanics where qubits will not suffice[54,55]. Now we have demonstrated

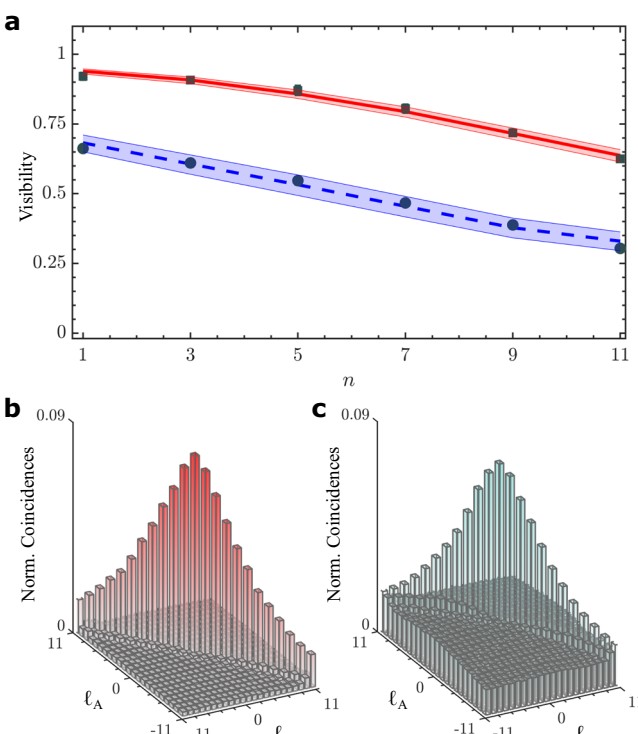

**Fig. 3 Experimental visibilities and modal spectrum. a** Visibility measurements for low (top solid line) and high noise (bottom dashed line) levels. The points are the experimental visibilities, while the lines correspond to the fitted values of dimension and purity. Measured spiral spectrum for the **b** low and **c** high noise levels. The shaded area corresponds to the uncertainty in the fit (standard deviation).

| Table 1 Purity and dimensionality measurements. | | | | | |
|---|---|---|---|---|---|
| Noise level | $p$ | $K$ | $Q$ | $\hat{p}$ | $\hat{K}$ |
| Low | 0.45 ± 0.03 | 22.84 ± 0.62 | 19.19 ± 0.22 | 0.44 ± 0.01 | 22 ± 1 |
| High | 0.13 ± 0.01 | 17.73 ± 0.71 | 3.76 ± 0.57 | 0.13 ± 0.02 | 18 ± 1 |

Measured purity ($p$) and dimensionality ($K$), under low and high noise levels, compared to estimates from other methods. Here, $Q$ is the average quantum contrast. Our experiment used a gating time of 25 ns for the coincidences with averaging over 10 s. Reducing the gating time, increasing the averaging time and taking care with the experimental conditions would significantly enhance the purities[29], even for the low noise conditions.

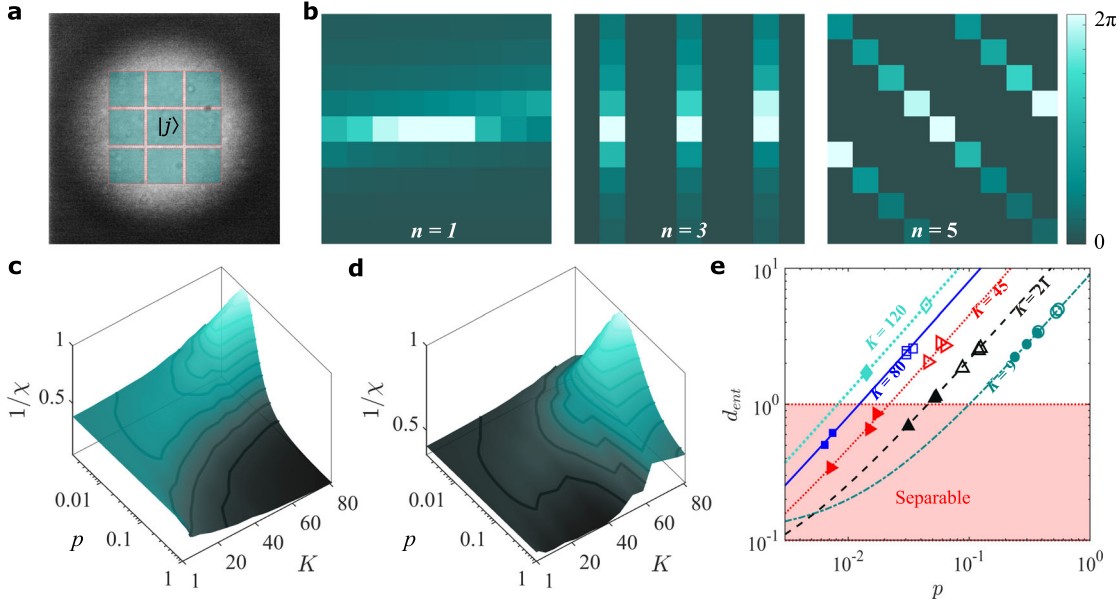

**Fig. 4 Demonstration using the pixel basis. a** The basis of our entanglement is expressed as pixels, illustrated here across the SPDC source (measured at the crystal), where each pixel is our state, $|j\rangle$. **b** Example holograms in this basis for the visibility measurements, shown here for $n = 1$, 3 and 5. **c, d** show the measurement outcomes for two example cases of dimension and purity, with and without background subtraction, respectively. All the measurements outcomes are plotted as data points in (**e**) showing the Schmidt rank, $d_{ent}$ as a function of purity, $p$, in excellent agreement with theory (lines). The dimensionality, $K$, is quoted for each case. Error bars are too small to be visualised.

a simple approach to return these crucial numbers. Although this paper is not a report on noise in high-dimensional quantum systems, we have deliberately introduced noise in order to demonstrate the robustness of the approach and to attain a range in purities for test purposes. The impact of noise in realising pure high-dimensional quantum states is only beginning to emerge[6], revealing that there are limits to the dimensionality that can be reached based on the quantum contrast and noise in the system. Our findings are entirely consistent with these reports. Although in the final test we have added noise (accidentals) subtraction to illustrate the juxtaposed position, this is strictly speaking not advisable[56].

Unlike a conventional Schmidt decomposition, we do not assume the state is pure, and the dimension extracted from our technique is conditioned on the presence of entanglement: a maximally mixed and maximally entangled system cannot yield the same result. While our approach would benefit from the knowledge of the modal spectrum, which can be measured very quickly[57,58], the outcome on purity and dimensionality are only modestly affected by typical spectrum shapes (see Supplementary Fig. 5); in our examples, the uncertainty in dimensionality is ≈5% with knowledge of the spectrum increasing to ≈10% without.

In addition to well-established quantum tomography tools, there are several new methods for characterising, measuring and extracting information about different quantum states[10–12]. Each method has its own set of pros and cons, regarding measurement time, the total number of required measurement settings and the ease of implementation. The method that we present here complements the existing techniques, providing one of the fastest mechanisms to extract and estimate valuable information about high-dimensional entangled quantum states.

The advantage of our method is the linear scaling to the number of measurements and the flexibility of the minimisation procedure. Our approach is thus an excellent candidate for a fast and easy test of purity and dimensionality prior to a more lengthy tomography, if necessary. One of the limitations of our technique is that prior knowledge of the form of the underlying state is

required for accurate fitting. Thus, we cannot provide definite proof of entanglement in an assumption-free manner.

However, the easy construction of our analysers and the resulting measurement of only visibilities reduces the complexity of characterising quantum states significantly when contrasted with QST-based approaches. Finally, our measurement approach has been tested against the topical isotropic state, but we point out that the construction of the analysers is not dependent on this state. This is analogous to other methods where extracting a measure always requires a target state, e.g., the fidelity from a QST measured against a maximally entangled state. We envisage that it may be possible to generalise the theory to extract key parameters from states other than just the isotropic state.

In summary, we have developed a simple yet powerful technique to measure the dimensionality and purity of high-dimensional entangled photonic quantum systems. Our approach is robust, fast, and provides quantitative values rather than bounds or witnesses, and works on both pure and mixed states. Our scheme exploits visibility in fringes after joint projections, making it fast and easy to implement, returning the key parameters of the system in a fraction of the time that a QST would take. Thus, we believe that our approach will be useful as a quick test with minimal experimental effort prior to more comprehensive state testing, valuable to the active research in high-dimensional spatial mode entanglement and foster its widespread deployment in quantum-based protocols.

## Methods

**High-dimensional state projections.** Our analysers project onto the high-dimensional Hilbert space, $\mathcal{H}_d$, mapping onto the states in Eq. (9), i.e., $|M, \alpha\rangle_n$, repeated here as

$$|M, \alpha\rangle_n = \mathcal{N} \sum_{j=0}^{d-1} c_{w_j,M}^n(\alpha)|j\rangle, \tag{5}$$

composed of coherent superpositions of basis states $|j\rangle \in \{|j\rangle, j = 0, 1..d-1\}$ with tuneable phases and amplitudes

$$c_{w_j,M}^n(\alpha) = e^{-i\pi w_j(n-1)/n} A_{w_j}^n c_{w_j,M}(\alpha), \tag{6}$$

and where $w_j = j - (d-1)/2$ and the factors

$$c_{w_j,M}(\alpha) = -\frac{ie^{-iw_j\alpha}}{\pi(M - w_j)}. \tag{7}$$

and

$$A_{w_j}^n = \begin{cases} 1, & \mathrm{mod}\{w_j, n\} = 0 \\ 0, & \text{otherwise} \end{cases}. \tag{8}$$

Here, $c_{w_k,M}(\alpha)$ controls the relative phases and amplitudes of the eigenmodes and $A_{w_j}^n$ modulates the coefficients' amplitudes while $\alpha \in [0, 2\pi/n]$. The spectrum given by Eq. (6) can be tuned by carefully selecting $n$, therefore enabling precise control of the subspaces that will be probed.

In the OAM basis, i.e., $|\ell\rangle \in \mathcal{H}_d$, the index $w_j$ can be replaced with the index $\ell \in \mathcal{Z}$. The mode projectors can be constructed from spiral phase profiles having the transmission function

$$U_n(\phi, \alpha) = \mathcal{M} \sum_{k=0}^{n-1} \exp(i\Phi_M(\phi; \beta_k \oplus \alpha)), \tag{9}$$

that is constructed from superpositions of fractional OAM modes[45,59],

$$\exp(i\Phi_M(\phi; \alpha)) = \begin{cases} e^{iM(2\pi+\phi-\alpha)} & 0 \le \phi < \alpha \\ e^{iM(\phi-\alpha)} & \alpha \le \phi < 2\pi \end{cases}, \tag{10}$$

having the same charge, $M$, but rotated by an angle $\beta_k \oplus \alpha = \mathrm{mod}\{\beta_k + \alpha, 2\pi\}$ for $\beta_k = \frac{2\pi}{n}k$, as illustrated in Fig. 1b, c for $n = 3$ and $n = 7$, respectively. Here, $\phi$ is the azimuthal coordinate and $\mathcal{M}$ a normalisation constant.

For the pixel basis, we constructed the holograms on a $d = D \times D$ grid with each square corresponding to a "pixel" state. The coefficients corresponding to a projection onto the state, $|M, \alpha\rangle_n$, can be mapped as

$$C_{r,c} = c_{w_{o-1},M}^n(\alpha), \tag{11}$$

where $o = (r-1)D - c$, for each index pair, $r, c = 1, 2, 3...d$, locating the row and column index of each pixel state on the grid. This mapping converts the list of coefficients, $c_{w_j}$ (for $j = 0, 1, ... d - 1$), into a square matrix $C_{r,c}$. To construct the hologram, we then extract the amplitude and phase of the matrix components of $C$ and obtain,

$$U_{r,c} = B_{r,c} \mod \{\arg(C_{r,c}), 2\pi\}, \tag{12}$$

where $B_{r,c} = |C_{r,c}|/\max(C)$. The final hologram can then be obtained by resampling $U$ onto a high-resolution grid that can be loaded onto the SLM. In this work, we resampled each projection hologram onto a $200 \times 200$ grid. Example holograms for $9 \times 9$ states that were resampled onto a $200 \times 200$ grid are shown in Fig. 4b.

**Experimental set-up**. The experimental set-up for the generation and measurement of entangled photons is illustrated schematically in Fig. 1d. A potassium-titanium-phosphate type I NLC was pumped with a 405 nm wavelength diode laser. The crystal temperature was set to obtain co-linear signal and idler entangled SPDC photons centred at a wavelength 810 nm. The photon pairs were then separated in path using a 50:50 beam splitter (BS). Each entangled photon was imaged onto an SLM using a 4f telescope ($f_1$ and $f_2$ having focal lengths of 100 and 500 mm, respectively), then subsequently coupled into single-mode fibres with a second 4f telescope (lenses $f_3$ and $f_4$ having focal lengths of 750 and 2 mm, respectively) and finally detected with avalanche photodetectors. Signals from each arm were measured in coincidences within a 25 ns coincidence window. The entangled photons were filtered with 10 nm bandpass filters centred at a wavelength of 810 nm. For our experimental demonstration, we restrict our measurements to a specific optical set-up and we varied the purity of the entangled state by introducing background noise in the form of white light. To obtain a high purity state ($p = 0.45$ in $K = 22$ dimensions), we had to reduce the laser power using a neutral density filter such as to reduce multiphoton emission events, which is known to have an impact on the purity of the SPDC photons[20]. To increase the noise in the system for the OAM basis measurements, we introduced background noise in the form of white light emitted by an incandescent light bulb until the quantum contrast (equivalently signal-to-noise ratio) dropped to 3. The measurement procedure of the quantum contrast is discussed in Supplementary Note 8.

**Optimal purity and dimensionality calculation**. Using the fact that the visibility obtained for each analyser is affected by the dimensionality and purity of the input state, we describe the procedure for determining their values for a given entangled quantum system, assuming it can be modelled by the isotropic state in Eq. (1). We measure the probability curves for $N$ analysers each with $n = 1, 3, ..., 2N - 1$, and compute their corresponding visibilities $V_n := V_n(p, K)$. This results in a set of $N$ non-linear equations that depend on the parameters $p$ and $K$. We then determine the optimal $(p, K)$ pair that best fit the function $V_n(p, K)$ to all $N$ measured visibilities by employing the method of least squares, which aims to minimise the

objective function

$$\chi^2(p, K) = \sum_{i=1}^{N} |V_{2i-1}^{\mathrm{The.}}(p, K) - V_{2i-1}^{\mathrm{Exp.}}|^2, \tag{13}$$

where the terms in the summation are the residuals (absolute errors) for each $n = 2i - 1$ visibility measurement (Exp.) with respect to the theory (The.).

## Data availability
The data that supports the plots within this paper and other findings of this study are available from the corresponding author upon reasonable request.

## Code availability
The codes that support the plots and multimedia files within this paper are available from the corresponding author upon reasonable request.

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

## Acknowledgements

We express our gratitude to Bienvenue Ndagano for his inputs. I.N. would like to acknowledge the Department of Science and Technology (South Africa) for funding. J.L. and F.Z. acknowledge support from the UK EPSRC (No. EP/T00097X/1). H.-C.H. acknowledges funding from the NTU Core Consortium project under Grant No. NTU-CC-109L892203.

## Author contributions

The experiment was performed by I.N. and V.R.-F., the theory was developed by I.N., F.Z., H-C.H. and J.L., the data analysis was performed by I.N., V.R.-F. and A.F. and the experiment was conceived by I.N., V.R.-F. and A.F. All authors contributed to the writing of the manuscript. A.F. supervised the project.

## Competing interests

The authors declare no competing interests.
