## [Peer Review File · Nature Communications]

Reviewers' Comments:

Reviewer #1:

Remarks to the Author:

The authors experimentally study isotropic states, entangled in orbital angular momentum. By performing conditional measurements, the authors manage to relate the visibility of the coincidence measurements (with single outcomes) to the inverse purity of the pure part of the isotropic state, as well as the degree of mixedness.

It is a diligently performed experiment that is well presented. Unfortunately, I cannot recommend the manuscript for publication in Nature communication in its current form for the following reasons. The authors motivate their study by introducing a way of determining high-dimensional entanglement for mixed states. Here, one should say that:

a) Such a measure exists, and is called Schmidt number (often referred to as entanglement dimensionality). It is confusing to also call the inverse purity of a pure state marginal Schmidt number, in my opinion, although I admit to have seen it in many quantum optics publications. While it is unfortunate that two different concepts with an equal name exist in such a narrow field, that would not be a reason for a rejection (there is no need for nominative dogmatism here). The problem I see is that the 'other' notion of Schmidt number already unambiguously and operationally captures the dimensionality of entanglement for every mixed state, not just the isotropic state. And for isotropic states there exist experimentally easy witnesses to detect it (for instance the state fidelity).

b) The authors capture the inverse marginal purity from a fit for the isotropic state. The problem here is that it will yield a nonzero entanglement dimension, even for completely separable states. The isotropic decomposition is only one of infinitely many ways of decomposing the density matrix. As the authors themselves note, the maximally entangled state with white noise becomes separable for $p=1/(d+1)$. With their fitting method, such values would still yield a nonzero K and thus 'measure' entanglement. Is that the intention? If yes, it needs to be better motivated. All that one can say about such a state is that the experimental data could have been obtained either by a mixture of separable states or by mixing a particular entangled state very strongly with white noise. What good is that information?

Both of these points can be verified in a number of publications, e.g. in a recent review by N. Friis et al Nat. Rev. Phys. 1, 72-87 (2019).

As a minor comment: to be consistent with the manuscript, the definition of K in the concept section should feature a 4th power instead of a square.

So that leaves us with a scheme that allows to infer purity and K for truncated isotropic states from cleverly chosen measurements. That in and of itself is not wrong and definitely a nice observation, masterfully executed in a top-notch experiment. But to rise to the level of scientific importance expected of Nature communications, I don't think that the proposed measure solves any critically outstanding problem and is highly unlikely to be used by the community in the future for the reasons explained above.

Reviewer #2:

Remarks to the Author:

The manuscript reports on an approach to certify the dimensionality and purity only with the requirement of simple projective measurements. The conventional method to reconstruct the density matrix is quantum state tomography, wherein the required number of measurements scales unfavourably with dimension. Later, Bavaresco et al., Nature Phys 14, 1032–1037 (2018) show how carefully constructed measurements in two bases are sufficient for certifying bipartite high-dimensional states and their entanglement. By using mode analysers constructed from the superposition of fractional OAM states, the authors show that the outcome returns a visibility that scales monotonically with entanglement dimensionality and visibility. The theory is clear and the experiments are also convincing. I think this manuscript could be considered for publication in Nature Communications.

I have a few comments regarding points that are slightly misleading in the present manuscript.

1. The dimension to be probed must be known a priori? In equation (3), the analysers are constructed on the $d=2L+1$ dimensional space.

2. I do not quite understand your statement as "The (perhaps) surprisingly low purity in our results is due to the fact that no background subtraction due to accidental counts was performed." The accidental counts can be directly estimated in experiment, why not perform the background subtraction?

3. In table I, the difference is not very obvious to me, In particular, " $K=22.84$ " and " $\hat{K}=22$ " have different number of significant digits.

4. In order to obtain the visibility, the coincidence count rates resulting from projections of biphotons onto the various analysing modes (determined by the value of θ) are registered. What's the required number for projective measurements? Is the number of trials much less than quantum state tomography and that approach presented in Nature Phys 14, 1032–1037 (2018) ?

Reviewer #3:

Remarks to the Author:

This work demonstrates a method to estimate the dimensionality and purity of high-dimensional states entangled in their OAM degree of freedom. Measurement of large quantum states is a challenging task due to rapid scaling in the number of projective measurements required for a full quantum state tomography. Various works investigate easier and quicker ways to estimate the amount of entanglement, dimensionality certification and witnessing, and so on.

Here, the authors perform a set of projective measurements on two entangled parties and observe interference fringes, from which the dimensionality and purity of the state can be estimated. An advantage of the method is the ability to project onto a state, formed by a superposition of different OAM values. This makes this approach differ a bit from the more common observation of Bell-type fringes.

The idea is definitely elegant and will be of interest to the specialists of quantum optics, but I hesitate to recommend it for Nature Communications. It does demonstrate an interesting way to analyze output from SPDC, but I think its limitations are significant enough for it not being an actual solution (although a step towards one) to the problem of dimensionality measurement of non-pure entangled states, as claimed in the article.

The method relies on the assumption that the analyzed state is a mixture of a pure state with an isotropic noise, a Werner-like form of Eq. (1). While this form is frequently used in literature for qubits, I would expect it to fail even within the example of an output of SPDC. One would expect a decrease in purity with the increase of OAM number l , due to increased noise in the measurement of large OAM states due to pixelization of SLMs, decrease in signal to noise ratio with the background due to lower amplitude, increased sensitivity to fluctuations, and so on. The pure part of the state is also assumed to be a superposition of many Bell states with different OAM, which is specific to SPDC output pumped by a Gaussian-shaped pump, but hardly covers the range of the useful high-dimensionally entangled states.

It is also unclear how this method compares with the other recent methods of estimation of high-dimensional entanglement, such as PRL 118, 110501 (2017) or Nat Comm 10, 2785 (2019) or other works including witnessing of genuine high-dimensional entanglement, PRL 120, 060502 (2018), PRA 97, 062309 (2018).

On a more technical side, parameter p from Eq. (1) is referred to as purity throughout the paper, which is incorrect. Purity by definition, as $\text{tr}(\rho^2)$, ranges from $1/d$ to 1, not 0 to 1 like the weight parameter p .

The experimental verification of the method, something the authors also note, shows an unreasonably low purity estimate. Multiple works, including some of the co-authors of this paper have shown much higher fidelities of high-dimensional entangled states coming out of SPDC. Multi-photon emission events are mentioned among the potential causes for extra noise. However, these events are specific only to SPDC pumped by high power pulsed lasers and do not affect the CW pumped process. Instead, the latter is more sensitive to the coincidence time setting, which is quite easy to adjust. The experiment in its current form is not very convincing that the method works in practice.

Responses to reviewers (NCOMMS-20-43280-T)

Reviewer #1:

The authors experimentally study isotropic states, entangled in orbital angular momentum. By performing conditional measurements, the authors manage to relate the visibility of the coincidence measurements (with single outcomes) to the inverse purity of the pure part of the isotropic state, as well as the degree of mixedness.

It is a diligently performed experiment that is well presented.

We thank the reviewer for the kind opening remarks. We hope to convince you that the work is truly worthy of publication in Nature Communications. We must add that, while the method was executed with OAM *as a topical example*, it is much more general. We have returned to the laboratory and added an additional example of the pixel basis, which we use to address some of your questions and improve the impact of the work.

Unfortunately, I cannot recommend the manuscript for publication in Nature communication in its current form for the following reasons. The authors motivate their study by introducing a way of determining high-dimensional entanglement for mixed states. Here, one should say that:

a) Such a measure exists, and is called Schmidt number (often referred to as entanglement dimensionality). It is confusing to also call the inverse purity of a pure state marginal Schmidt number, in my opinion, although I admit to have seen it in many quantum optics publications. While it is unfortunate that two different concepts with an equal name exist in such a narrow field, that would not be a reason for a rejection (there is no need for nominative dogmatism here). The problem I see is that the 'other' notion of Schmidt number already unambiguously and operationally captures the dimensionality of entanglement for every mixed state, not just the isotropic state. And for isotropic states there exist experimentally easy witnesses to detect it (for instance the state fidelity).

b) The authors capture the inverse marginal purity from a fit for the isotropic state. The problem here is that it will yield a nonzero entanglement dimension, even for completely separable states. The isotropic decomposition is only one of infinitely many ways of decomposing the density matrix. As the authors themselves note, the maximally entangled state with white noise becomes separable for $p=1/(d+1)$. With their fitting method, such values would still yield a nonzero K and thus 'measure' entanglement. Is that the intention? If yes, it needs to be better motivated. All that one can say about such a state is that the experimental data could have been obtained either by a mixture of separable states or by mixing a particular entangled state very strongly with white noise. What good is that information?

Both of these points can be verified in a number of publications, e.g. in a recent review by N. Friis et al Nat. Rev. Phys. 1, 72-87 (2019).

In our opinion parts (a) and (b) are coupled, so we will treat them as such. The pertinent point here is the difference between the approach to get the information out (the measurement), and what you then do with that information, e.g., calculating various measures. Perhaps our language was a little clumsy as what we are offering here is a measurement approach that provides important quantitative information, with few assumptions, and from which many typical quantities can be inferred. Further, the role of "noise" in a quantum system has become rather topical recently and seemed to be the best system to apply our method to, given that we believe it to be useful in practical applications in a real-world setting, hence the use of the isotropic state. Let us be specific to what we can do with the information from our measurements.

We are aware that the Schmidt rank, based on bounds of the fidelity with respect to some target state, already works for a wide class of quantum states [1]. We used the alternative version since it is perfectly valid and frequently used for characterising dimensions for pure states with respect to the eigenvalues extracted from the Schmidt decomposition, and now common in our field. But the information we extract from our approach is sufficient to also calculate the measure you mention: one can relate the pure state dimensionality (K) and purity (p) in our experiment to the fidelity of the state which can be used to extract

the Schmidt rank. This is because the visibility measurements in our technique are very sensitive to the dimensionality and purity and likewise can be used to extract information about the same system.

While a witness for the isotropic state indeed exists and has been discussed in detail in seminal works [1, 2], there are no direct methods to extract the “purity”, p , of a state. Our approach offers direct measurements of such quantities, and avoids the long route of quantum state tomography (QST), which indeed is the most complete method to use for a complete characterization of the quantum system, and also gives quantitative outcomes as ours, but scales unfavourably with dimension. Our approach scales linearly with dimension, making it highly practical for studying high-dimensional states (see later).

Our method returns the dimensionality of the pure component of the isotropic state when using K , as well as purity, even if the system is below the separability limit. But these combined can yield information about the entanglement dimensionality as used in the ref. [1]. Moreover, using the visibilities extracted from our technique, the problem is easily ameliorated by constraining the solution space, (p, K) , outside the region of separable states, $p > 1/(K + 1)$. To motivate the use of these constraints, we show below a plot of visibility vs p for several dimensions. For each analyser (marked by n), we see that there is a minimum visibility below which one can expect separability ($p \leq 1/(K + 1)$). Above this, one finds visibilities that can be explained by an entangled state. Accordingly, this means that we can rule out that part of the parameter space that would yield separable states. **We have adapted the relevant figures to show this.**

The fact that our technique can still detect correlations even below the separability criterion for isotropic states means that it is sensitive to correlations even in extremely high noise situations. This can be useful in situations where high resolution in quantum imaging is required although background noise levels are too high, e.g., in ghost imaging [3]. In the context of imaging, we will later discuss new experiments using pixels as a basis.

For the case where the system is an ensemble of separable states, the conditional visibilities, which strongly depend on non-local correlations between the photons, return a visibility of $V = 0$ for mixtures of separable states. To illustrate this point, consider the mixture of d , pure-states $|m\rangle|m\rangle$ with probabilities p_m and no coherent off-diagonal terms. For each analyser, the visibility will be $V = 0$. This implies that there are no high dimensional entanglement correlations. **We have added a worked-out example of this case in the supplementary material.**

Inspired by Fig. S2 from the supplementary material of ref. [1], we used our measurement data to calculate the Schmidt Rank, which is shown in [1] to be equivalent to $d_{ent} = d \times F$ where d is the global dimensions while F is the fidelity of the state being analysed with respect to a maximally entangled state. **We show simulation results for our measured dimension (K), the calculated dimensions using the fidelity ($d_{ent} = d \times F$) as well as the lower bound on dimensions with respect to p in new SI.**

Since our method extracts both p and K , we can approximate the fidelity of the state as $F_p = \frac{p(K^2-1)+1}{K^2}$. We can therefore compute the Schmidt Rank from $d_{ent} = K \times F_p$. In the graph above we run simulated experiments and compute this parameter using our measurement approach and that of [1]. We can see that the agreement is very good. This means that our method can also approximate the fidelity of the state with respect to maximally entangled state in the presence of noise, and deduce the alternative dimensionality definition. The shaded region corresponds to $p \leq \frac{1}{K+1}$ where no entanglement is considered.

While our technique was demonstrated with the isotropic state as a highly relevant and topical case, we anticipate that it can be extended to other forms of entanglement, something worth investigating in future work. We acknowledge this in the paper.

To summarize the above:

- All separable states can be ruled out by our method since the conditional visibility measurements only yield non-zero values for completely separable states (ensembles of separable states). For the isotropic state, restrictions on p can be used to constrain the fitting procedure, i.e., only search within the range $\frac{1}{K+1} < p \leq 1$.
- Since our technique extracts the dimensions (K) of pure state with a probability of p , this can be used to calculate the Schmidt Rank as defined in [1], by approximating the fidelity (F_p) using the measured K and p .
- The message is that we have sufficient information from our approach to infer the value for the standard measures the community may wish to adopt.

[1] Bavaresco, J., Valencia, N.H., Klöckl, C., Pivoluska, M., Erker, P., Friis, N., Malik, M. and Huber, M., 2018. Measurements in two bases are sufficient for certifying high-dimensional entanglement. *Nature Physics*, 14(10), pp.1032-1037.

[2] Ecker, S., Bouchard, F., Bulla, L., Brandt, F., Kohout, O., Steinlechner, F., Fickler, R., Malik, M., Guryanova, Y., Ursin, R. and Huber, M., 2019. Overcoming noise in entanglement distribution. *Physical Review X*, 9(4), p.041042.

[3] Gregory, T., Moreau, P.A., Toninelli, E. and Padgett, M.J., 2020. Imaging through noise with quantum illumination. *Science advances*, 6(6), p.eaay2652.

To address these issues we have taken the following steps:

- We have altered the title and explanations to reflect the subtle issue of measurement approach and what one can extract from the information, the choice of measure;

- We have added constraints on non-separability to all figures (where appropriate);
- We have incorporated alternative measures such as the Schmidt Rank, d_{ent} , into our measurement outcome analysis, and discussed in detail the role of noise and its importance for high-dimensional state quantification;
- We have added new supplementary information that explains all the relations, and shows some simulated examples;
- Most notably, we have performed new experiments in the pixel basis and used those examples to frame our work in the context of the measures mentioned by the referee, and to highlight the impact of noise on the outcomes. The referee will see that we provide experimental data for the Schmidt Rank in the new figure 4.

As a minor comment: to be consistent with the manuscript, the definition of K in the concept section should feature a 4th power instead of a square.

Thank you, this was a typo on our side. All equations have now been carefully checked.

So that leaves us with a scheme that allows to infer purity and K for truncated isotropic states from cleverly chosen measurements. That in and of itself is not wrong and definitely a nice observation, masterfully executed in a top-notch experiment. But to rise to the level of scientific importance expected of Nature communications, I don't think that the proposed measure solves any critically outstanding problem and is highly unlikely to be used by the community in the future for the reasons explained above.

We hope that we have now addressed the issue of “measure” and “measurement”, and with the new experimental data shown that we can go beyond just purity and K, and instead use this to also infer fidelity and Schmidt rank. What is the advantage? We believe it is the quantitiveness of our approach, giving values rather than bounds for a variety of information, its versatility to be used across many basis types, ease of implementation through simple holograms (no adjustment for hologram efficiencies needed), and speed.

So far we have not really concentrated on the speed of measurement. The number of measurements in our technique scales linearly with d whereas full QST (which also gives quantitative measurements) scales as d^4 and the reduced QST (with two MUBs) which returns bounds, scales as d^2 . To take an example, our approach takes a few mins for a ~ 100 dimensional test, whereas a QST would take a couple of decades and the reduced QST takes on the order of a day. Moreover, since the projections are constructed from simple superpositions and only visibilities are recorded, the complexity is significantly reduced since our technique does not require any special basis construction and no post measurement adjustment to the outcomes due to efficiency variation from complex amplitude modulation on holograms. We can also test in any dimension, and not only those where MUBs are known. What this means to the user is that when selecting which dimension to test in, there is no significant penalty for choosing 100 dimensions over 10 dimensions (say) – both yield quantitative results in minutes. The same is not true for any other approach. We think this is why the work is likely to be adopted by the community as an important tool.

To illustrate this, we returned to the laboratory and applied our approach to the pixel basis, in >100 dimensions, extracting the key information of purity, dimension and Schmidt Rank (as defined and measured in other works), all for mixed states in the presence of noise.

We hope that this comprehensive response and new additions to the main text and SI addresses all the concerns of the referee. We hope that it will sway you to believe that the work is highly topical (high-dimensional states in noise is still very new), an important advance in the toolkit of high-dimensional quantum state analysis, and will appeal to the wide audience of this journal.

Reviewer #2:

The manuscript reports on an approach to certify the dimensionality and purity only with the requirement of simple projective measurements. The conventional method to reconstruct the density matrix is quantum state tomography, wherein the required number of measurements scales unfavourably with dimension.

Later, Bavaresco et al., Nature Phys 14, 1032–1037 (2018) show how carefully constructed measurements in two bases are sufficient for certifying bipartite high-dimensional states and their entanglement. By using mode analysers constructed from the superposition of fractional OAM states, the authors show that the outcome returns a visibility that scales monotonically with entanglement dimensionality and visibility. The theory is clear and the experiments are also convincing. I think this manuscript could be considered for publication in Nature Communications.

We thank the reviewer for their comments and positive remarks. This is a good summary of the work. We now returned to the laboratory and added new data to show that the work is not specific to OAM but far more general.

I have a few comments regarding points that are slightly misleading in the present manuscript.

1. The dimension to be probed must be known a priori? In equation (3), the analysers are constructed on the $d=2L+1$ dimensional space.

No, it does not need to be known. One must always select a test space though (for all techniques), and the dimensionality of the test space can be arbitrarily chosen, i.e., any value of d . You test for entanglement in some space, and we will tell you how many dimensions are entangled there. The “size” of the Hilbert space one wants to search through can be as large as one pleases. It is analogous to making sure that the camera you use to take an image is big enough to fit that image on. The pertinent point is that there is no significant penalty for going “too big” in your test dimension, and this differentiates our approach over others – the speed of measurement. Selecting 100 dimensions over 10 dimensions to test in changes the measurement time from “a minute” to a “few minutes”. Using QST, the other approach to returning quantitative outcomes, such a decision would alter your measurement time from “several hours” to a “couple of decades” – i.e., you simply could not do this. Our approach is unique in this regard.

We have tidied up the language and notation to make this clearer, removing “L” altogether.

2. I do not quite understand your statement as “The (perhaps) surprisingly low purity in our results is due to the fact that no background subtraction due to accidental counts was performed.” The accidental counts can be directly estimated in experiment, why not perform the background subtraction?

We thank the reviewer for bringing this up. The topic of noise and its impact on high-dimensional states is very much in its infancy, with some surprising consequences. For example, there are limits to what can actually be achieved based on the quantum contrast of the experiment – see for example the attached figure from [<https://arxiv.org/pdf/1908.08943.pdf>].

FIG. 1. The relationship between Q , d and p from the isotropic state. The white lines indicate constant values of p . We see that maintaining a constant value of p as d increases requires an increase to the signal-to-noise ratio Q .

In all our prior work we too have subtracted background counts, then finding very high purity states. Here we were motivated not to do this primarily by two factors: (1) we wanted to have a range of purities to convince the reader that the approach works for a range of states, so we deliberately did not subtract

backgrounds and even increased the light/source levels to get the range, and (2) we wanted to illustrate the method in the presence of noise, a topical subject [1-3]. But just to be clear, the work isn't about reporting on noise and its impact, but rather using deliberate and accidental noise as a means to show a range of measureable purities and dimensions as a form of validation.

There is in fact a third reason why one should not subtract background counts in general. This kind of "artificial" noise subtraction makes certain assumptions about the detectors, environment, and photon statistics of the source; that the state coming from the source is pure and that the detectors and environment are noisy. At times, these assumptions can be incorrect since the multiphoton emissions of the source can be a main culprit [1]. The use of background-subtracted measurements does not meet the strict requirements for most fundamental tests of entanglement and should be at all costs avoided since it introduces loopholes, for example, in a Bell-inequality test [2].

But we do appreciate the confusion. In the revised manuscript we have added significant text to make this clear, we have removed the statement about "surprise" (actually only the corresponding author was surprised at the impact on the dimensionality) and also show some new data with and without background subtraction in up to >100 dimensions, adding a statement as to why one may not want to do the latter in practice. We also add a table to the SI to show the experimental conditions that we used to change the purity.

[1] Xu, P., Yong, H.L., Chen, L.K., Liu, C., Xiang, T., Yao, X.C., Lu, H., Li, Z.D., Liu, N.L., Li, L. and Yang, T., 2017. Two-hierarchy entanglement swapping for a linear optical quantum repeater. *Physical Review Letters*, 119(17), p.170502.

[2] Larsson, J.Å., 2014. Loopholes in Bell inequality tests of local realism. *Journal of Physics A: Mathematical and Theoretical*, 47(42), p.424003.

[3] Ecker, S., Bouchard, F., Bulla, L., Brandt, F., Kohout, O., Steinlechner, F., Fickler, R., Malik, M., Guryanova, Y., Ursin, R. and Huber, M., 2019. Overcoming noise in entanglement distribution. *Physical Review X*, 9(4), p.041042.

3. In table I, the difference is not very obvious to me, In particular, " $K=22.84$ " and " $\hat{K}=22$ " have different number of significant digits.

Thank you, we have fixed the significant figures. The values with the $\hat{}$ were extracted through other approaches, but with the significant disadvantage to get \hat{p} you needed to already know \hat{K} , and vice versa. Whereas our measured K and p did not require any prior knowledge of the other.

4. In order to obtain the visibility, the coincidence count rates resulting from projections of biphotons onto the various analysing modes (determined by the value of θ) are registered. What's the required number for projective measurements? Is the number of trials much less than quantum state tomography and that approach presented in Nature Phys 14, 1032–1037 (2018) ?

This is an excellent question. The number of measurements in our technique scales linearly with d whereas full QST (which also gives quantitative measurements) scales as d^4 and the reduced QST (with two MUBs – the paper you mention) scales as d^2 . To take an example, our approach takes a few mins for a ~ 100 dimensional test, whereas a QST would take a couple of decades and the reduced QST takes on the order of a day. Moreover, since the projections are constructed from simple superpositions and only visibilities are recorded, the complexity is significantly reduced since our technique does not require any special basis construction and no post measurement adjustment to the outcomes due to efficiency variation from complex amplitude modulation on holograms. We can also test in any dimension, and not only those where MUBs are known. What this means to the user is that when selecting which dimension to test in, there is no significant penalty for choosing 100 dimensions over 10 dimensions (say) – both yield quantitative results in

minutes. The same is not true for any other approach. We think this is why the work is likely to be adopted by the community.

To illustrate this, we returned to the laboratory and applied our approach to the pixel basis, in >100 dimensions. We have also added a discussion on this point in the main text.

Reviewer #3:

This work demonstrates a method to estimate the dimensionality and purity of high-dimensional states entangled in their OAM degree of freedom. Measurement of large quantum states is a challenging task due to rapid scaling in the number of projective measurements required for a full quantum state tomography. Various works investigate easier and quicker ways to estimate the amount of entanglement, dimensionality certification and witnessing, and so on.

Here, the authors perform a set of projective measurements on two entangled parties and observe interference fringes, from which the dimensionality and purity of the state can be estimated. An advantage of the method is the ability to project onto a state, formed by a superposition of different OAM values. This makes this approach differ a bit from the more common observation of Bell-type fringes.

The idea is definitely elegant and will be of interest to the specialists of quantum optics, but I hesitate to recommend it for Nature Communications. It does demonstrate an interesting way to analyse output from SPDC, but I think its limitations are significant enough for it not being an actual solution (although a step towards one) to the problem of dimensionality measurement of non-pure entangled states, as claimed in the article.

We thank the reviewer for the positive comments. Indeed the approach is elegant and fast, and we hope it will be of interest to a very wide audience. In particular, our approach can be generalised to any quantum entangled system since our approach returns sufficient information to infer properties of general states, e.g., Fidelity and Schmidt Rank. This is because the visibilities carry information about the state. As for the significance of the approach, please see our response to referees #1 and #2, where we clearly show that our technique requires far less measurements than the present state of the art.

We only used SPDC as an example source since it is a well-studied phenomenon and is used in a myriad of quantum information protocols – our approach is not limited to this. Nor is it limited to OAM. Our decomposition of the density matrix into two components allows for the pure state $|\psi\rangle$ to be anything. Remember that the representation of the state in the Schmidt basis is what is important here and is even independent of the degree of freedom, i.e., any degree of freedom can be written in a complete basis and that is all that one requires to apply our method. Crucially, the method can be extended to any degree of freedom provided the Schmidt decomposition is known, meaning that the method does not rely on a particular kind of state.

To highlight these points, we have returned to the laboratory and shown new data in the pixel basis, in dimensions in excess of 100. We have significantly improved the results and discussion to reflect on the many advantages that this approach brings.

The method relies on the assumption that the analysed state is a mixture of a pure state with an isotropic noise, a Werner-like form of Eq. (1). While this form is frequently used in literature for qubits, I would expect it to fail even within the example of an output of SPDC. One would expect a decrease in purity with the increase of OAM number l , due to increased noise in the measurement of large OAM states due to pixelization of SLMs, decrease in signal to noise ratio with the background due to lower amplitude, increased sensitivity to fluctuations, and so on. The pure part of the state is also assumed to be a superposition of many Bell states with different OAM, which is specific to SPDC output pumped by a Gaussian-shaped pump, but hardly covers the range of the useful high-dimensionally entangled states.

The reviewer has highlighted a crucial point. The Werner “isotropic state” is not only used to study qubits states, it can be used for modelling high dimensional quantum states in the presence of noise, as has been published recently in [1], where these states have been showcased as a main model for demonstrating high-

dimensional state robustness in [2] and [3] for photon pairs and multi-photon states, respective. Moreover, reviewer 1 also mentioned that such modes have “witnesses” specifically designed for them.

As for the introduction of noise due to SLM pixelation, we believe that one can overcome such limitations by working within its limits. Specifically, for given SLM specifications, it is possible to determine the largest OAM mode to encode on the SLM whose fidelity to the theoretical mode is high enough to consider it a good approximation to it [4]. Nonetheless, this has been thoroughly treated in [1] (and in the pixel basis [5]) and it has been shown that the high tolerance of entanglement to noise is still prevalent even under such conditions.

On the final point (please also see the previous answer): While it is true that the state is a superposition of many Bell states, it is still a valid Schmidt decomposition on the higher-dimensional Hilbert space. The only difference is that the SPDC mode field modulates the coefficients of the contributing eigenmodes. Any entangled state generally has the same form (Schmidt decomposition) unless it has some degree of mixture. In addition, one can switch the OAM modes with any other degree of freedom, i.e., time-energy bins and position, and while we use SPDC for convenience in our experiments, the theory does not assume an SPDC source, nor OAM, nor Gaussian pumps, and so on. This is primarily because the mode indexes are just labels and be replaced to suit any degree of freedom. The approach is very general, as one can see from the new text around Eq. (1).

To illustrate this point, we have adjusted the concept text substantially, have added new simulations to the SI, and more importantly, we have performed new experiments to highlight the versatility. Using the pixel basis, we have shown results with deliberate noise and with noise subtraction (but see comments to referee #2), for >100 dimensions.

Furthermore, we anticipate that it can be extended to other forms of entanglement beyond the isotropic state, something worth investigating in future work.

We have added discussions on these points in the main text.

[1] Ecker, S., Bouchard, F., Bulla, L., Brandt, F., Kohout, O., Steinlechner, F., Fickler, R., Malik, M., Guryanova, Y., Ursin, R. and Huber, M., 2019. Overcoming noise in entanglement distribution. *Physical Review X*, 9(4), p.041042.

[2] Bavaresco, J., Valencia, N.H., Klöckl, C., Pivoluska, M., Erker, P., Friis, N., Malik, M. and Huber, M., 2018. Measurements in two bases are sufficient for certifying high-dimensional entanglement. *Nature Physics*, 14(10), pp.1032-1037.

[3] Huber, M., Mintert, F., Gabriel, A. and Hiesmayr, B.C., 2010. Detection of high-dimensional genuine multipartite entanglement of mixed states. *Physical review letters*, 104(21), p.210501.

[4] Pinnell, J., Rodríguez-Fajardo, V. and Forbes, A., 2020. Probing the limits of orbital angular momentum generation and detection with spatial light modulators. *Journal of Optics*, 23(1), p.015602.

[5] Valencia, N.H., Srivastav, V., Pivoluska, M., Huber, M., Friis, N., McCutcheon, W. and Malik, M., 2020. High-dimensional pixel entanglement: efficient generation and certification. *Quantum*, 4, p.376.

It is also unclear how this method compares with the other recent methods of estimation of high-dimensional entanglement, such as PRL 118,110501 (2017) or Nat Comm 10, 2785 (2019) or other works including witnessing of genuine high-dimensional entanglement, PRL 120, 060502 (2018), PRA 97, 062309 (2018).

We point the reviewer to our last answer to reviewer 1 (a) and (b). The methods above solve a different problem, witnessing entanglement in high dimensions. This means that such techniques use various parameters, e.g., the entanglement of formation, to quantify high strong the correlations in a d dimensional state and as highlighted in (Nat Comm 10, 2785 (2019)) work well if the state is nearly “maximally

entangled". While in our case, we measure key parameters of a high-dimensional entangled system and give quantitative values for the dimension and purity, two parameters of utmost importance for quantum information processing and communication. Moreover, those techniques require measurements with MuBs for high accuracy, which can be difficult to construct and whose number of measurements scales unfavourably with increasing dimensions. Reduced approaches use partial information from the density matrix to make estimations on the entanglement of formation (EoF), or to place bounds on the parameters. Our approach is fast and quantitative.

A fair comparison of our work is to the current state of the art [1], which requires measurements that scale as $\sim d^2$, returning bounds only. Our scheme can extract the important parameters quantitatively, with measurements that scale linearly with dimension, d , and adding in extra key information such as the purity of the state. This significantly advances our toolkit for charactering high dimensional quantum states without a complete tomography.

[1] Bavaresco, J., Valencia, N.H., Klöckl, C., Pivoluska, M., Erker, P., Friis, N., Malik, M. and Huber, M., 2018. Measurements in two bases are sufficient for certifying high-dimensional entanglement. *Nature Physics*, 14(10), pp.1032-1037.

On a more technical side, parameter p from Eq. (1) is referred to as purity throughout the paper, which is incorrect. Purity by definition, as $\text{tr}(\rho^2)$, ranges from $1/d$ to 1, not 0 to 1 like the weight parameter p .

The reviewer is correct. Nevertheless, we believe that it is still correct to do so since a perfectly separable state has $p \leq \frac{1}{d+1} \sim \frac{1}{d}$ for dimensions d , and $p = 1$ for a pure state. Thus, in a practical sense these definitions are approximately equivalent, particularly for high-dimensional states which is the subject of our paper, and therefore we believe it is correct to use them interchangeably.

But we agree this should be spelt out, so we have clearly motivated the use of this term in the text.

The experimental verification of the method, something the authors also note, shows an unreasonably low purity estimate. Multiple works, including some of the co-authors of this paper have shown much higher fidelities of high-dimensional entangled states coming out of SPDC. Multi-photon emission events are mentioned among the potential causes for extra noise. However, these events are specific only to SPDC pumped by high power pulsed lasers and do not affect the CW pumped process. Instead, the latter is more sensitive to the coincidence time setting, which is quite easy to adjust. The experiment in its current form is not very convincing that the method works in practice.

We thank the reviewer for raising this point; it is the same question asked by referee #2 and so we ask that you read the full answer there. In summary, we deliberately make the noise levels high to invoke low purities just to have a spread of conditions. It is not a reflection of the experiment poorly executed. It is very common

in the quantum structured light community to attain high purities by accidental subtraction, but the impact of this decision, and indeed the impact of noise in general, is only beginning to be understood.

FIG. 4. (Color online) (a) Linear entropy and (b) fidelity as a function of dimension. The error for both of these measurements is ± 0.01 , which is too small to be seen clearly on the graphs. In each case, the squares represent the measured data, while the circles represent the threshold states in Eq. (13). The shaded area represents the set of states that will not violate the appropriate high-dimensional Bell inequality.

While it may be true that the CW SPDC process may have minor photon contributions, the projection measurements (SLM) also introduce noise due to pixelation. This implies that, as one increases the dimensions of a quantum systems, more noise creeps in due to the growing size of the Hilbert space ([2] and [3]). This can also be seen in Fig. 4 from ref [4] (shown below), where the fidelity of a quantum system relative to a maximally entangled state decays while the linear entropy increases with increasing entanglement dimensions. This substantiates our findings.

But we do appreciate the confusion. **In the revised manuscript we have added significant text to make this clear, we have removed the statement about “surprise” (actually only the corresponding author was surprised at the impact on the dimensionality) and also show some new data with and without background subtraction in up to >100 dimensions, adding a statement as to why one may not want to do the latter in practice. We also add a table to the SI to show the experimental conditions that we used to change the purity.**

[1] Larsson, J.Å., 2014. Loopholes in Bell inequality tests of local realism. *Journal of Physics A: Mathematical and Theoretical*, 47(42), p.424003.

[2] Zhu, F., Tyler, M., Valencia, N.H., Malik, M. and Leach, J., 2019. Is high-dimensional photonic entanglement robust to noise?. *arXiv preprint arXiv:1908.08943*.

[3] Ecker, S., Bouchard, F., Bulla, L., Brandt, F., Kohout, O., Steinlechner, F., Fickler, R., Malik, M., Guryanova, Y., Ursin, R. and Huber, M., 2019. Overcoming noise in entanglement distribution. *Physical Review X*, 9(4), p.041042.

[4] Agnew, M., Leach, J., McLaren, M., Roux, F.S. and Boyd, R.W., 2011. Tomography of the quantum state of photons entangled in high dimensions. *Physical Review A*, 84(6), p.062101.

Reviewers' Comments:

Reviewer #1:

Remarks to the Author:

I want to thank the authors for taking my comments seriously and investing extra effort in presentation and even experiment to address most of them. I think the article has greatly improved and I would be happy to see it published eventually.

I am still not entirely certain that one of the central issues that made me hesitant to recommend the article is resolved, but it is somewhat inherent to the proposal and I don't think extra revisions could solve it.

The remaining point I speak of is the intrinsic assumption that is used for the entire experiment. Instead of treating general noise, the authors still need to assume that the state has a specific form, that of an isotropic state. It is inherent in the method, as it is exactly the goal of the authors to fit p (the isotropic noise parameter) through minimal experimental effort. This will always limit the method to be an estimation technique, and I don't see how one can practically certify entanglement (or its dimension) using such assumptions. The advantage of witnesses would be that they can directly be used as certificates for adversarial protocols, as they do not make assumptions about the source/channels (i.e. states). That being said, having a quick first estimate, before proceeding to more involved techniques has its merits. And the authors are right that this first estimate compares well against quantum state tomography. But what about direct measurements of fidelity (scales as d) or even just two unbiased measurements? Is it really faster?

As a side-remark, I think the authors did well in also estimating the Schmidt number. I don't agree that, as the authors write 'K reveals the entangled dimensions ...'. K reveals the inverse purity. It happens to be 1 for separable states and d for states entangled in all dimensions, but in-between there is no operational connection between the statement from the authors and the meaning of that number. There are states where all dimensions certifyably contribute to the entanglement, yet K is close to 1. Due to the monotonicity of Renyi entropies, I believe that K is a lower bound to the Schmidt rank and thus dimensionality, but it can't be reasonably equated to 'entangled dimensions' more than any other entropy (e.g. why $\text{Tr}(\rho^2)$? Why not $\text{Tr}(\rho^3)$? They all have a similar merit in quantifying entanglement).

My conclusion is that is a great experiment, good and honest science, but my prediction is that the method makes too many assumptions to likely be adopted. But I would also be happy to be proven wrong and ultimately only time will tell.

Reviewer #2:

Remarks to the Author:

I am impressed by the diligent work done by the authors. I feel that all of my concerns have been well addressed in the response Letter and in the revision. Thus I would like to recommend its publication in Nature Communications.

Reviewer #3:

Remarks to the Author:

This manuscript has certainly improved after a revision, especially new data reporting on a measurement in a different type of spatial basis (coordinate vs OAM). I now could imagine the concept and the measurement idea, which I find interesting and elegant, to be of interest for the broad reader audience of Nature Communications, although I doubt it will be a widely adopted method of quantum state characterization.

The purities reported in the new results are again uncharacteristically low when compared to similar measurements, including a number of those that do not subtract background noise.

Comparing, for example, with the pixel-entanglement work mentioned in the response [Quantum 4, 376 (2020)], where state fidelities of 95% are reported for dimensions of up to 20, while the new data of the manuscript reports purities (and fidelities via Eq.40) that are essentially almost zero for anything with a dimension above 9. Thus, the demonstration does show qualitative dependence but fails to perform as a quantitative measurement.

I would like to reiterate, that part of the problem, at least in the OAM case lies in the fact that this measured state cannot be modeled accurately as a Werner-like state with isotropic noise. The amount of observed/measured noise is higher for the parts of the state formed by larger OAM values. This is a well-known fact, demonstrated multiple times since a long time ago including by some of the co-authors of this manuscript [New J. Phys. 11 103024 (2009)].

Finally, given the recent advances in adaptive and self-guided tomography [PRL 117, 040402 (2016); PRA 101, 022317 (2020); PRL 126, 100402 (2021)], the protocol of Bavaresco et al. for dimensionality witnessing, and the textbook SWAP test [Opt. Express 26, 8443 (2018); arXiv:2103.10219], for direct measurement of the purity of any quantum state, I find the scaling comparison with the standard QST to be forced.

REVIEWER COMMENTS

General Response to the reviewers

All reviewers seem to be happy with the quality of the work, mentioning that the approach is elegant, interesting and novel, the experiments excellently done, the additional versatility shown by the pixel basis examples compelling, and the topic interesting and relevant to the audience of this journal. We are very encouraged by these positive remarks. Our original goal was to extract, in a fast and simple manner, the purity and dimensionality, mixed or pure, expressed in any basis, in high dimensions, while contending with the topical issue of noise. This we have done, and forms the central message of our manuscript. One can think of our approach, in a colloquial sense, as a happy marriage between Bell-type measurements and tomography-type measurements. Our approach is quantitative, fast, works for any basis and any dimension, and from the measurements one can quickly infer other important measures (such as Fidelity). Because the recipe is simple and fast, we believe it makes for an excellent first test to extract the important parameters. We think these benefits make it suitable for this journal.

The main criticism is that this only works for the isotropic state, but here we wish to make some important points: (1) the measurement approach we are proposing here is not state dependent – at no stage in the construction of the projectors or the measurement steps is the “isotropic state” encoded. (2) It is indeed so that we use the isotropic state as our test example; as we have explained, this is because it covers very practical situations, and incorporates the highly topical and important issue of noise and its impact. Our measurement outcomes allows us to extract the important parameters quantitatively for this important case, in particular the purity which is usually inferred from a density matrix construction. It is not a trivial or special case – it is a well-known universally applied example covering many, many practical situations. Most approaches make some such assumptions about what you try to extract after the measurements, e.g., Fidelities from QSTs are calculated with respect to some target state (often maximally entangled), and so on. This is not so unusual. What is important is that the measurement does not assume this, and neither does ours. (3) This then begs the question: do the visibilities in our approach hold enough information to extract the key parameters for other target states? There is no reason why it certainly cannot work, i.e., not a definite “no”. Objectively then, it is possible that the approach is applicable beyond even the general case we consider. After all, it is the want of this journal to provide a route to new applications and studies. One could say (with a little generosity) that we have demonstrated it for the isotropic state, but not that it only works for this state.

To summarise the benefits:

- Our method produces visibilities that hold key information about the state, including dimensions, purity, fidelity and so on, all of which can be extracted quantitatively.
- The projections are intuitive and easy to construct, are adaptable to any degree of freedom, and not restricted in dimension, e.g., work for prime and non-prime dimensions alike, and their construction is not dependent on the state under test.

- Our approach is very fast, scales linearly with dimensions, making it highly valuable for a fast test of the state, regardless of its dimension, returning results “in minutes”.
- Our approach has been shown to work for the highly topical case of noisy systems, and has the potential to be adapted to other situations, e.g., lossy systems, modal cross-talk and so on. This is because the measurement approach and calculating the measures are decoupled.

We hope that in this latest reply and revision we can convince you that these benefits and the quality level of the work make it suitable for this journal.

Reviewer #1 (Remarks to the Author):

I want to thank the authors for taking my comments seriously and investing extra effort in presentation and even experiment to address most of them. I think the article has greatly improved and I would be happy to see it published eventually.

We thank the reviewer for the very positive remarks. We do like the final form the manuscript is taking.

I am still not entirely certain that one of the central issues that made me hesitant to recommend the article is resolved, but it is somewhat inherent to the proposal and I don't think extra revisions could solve it. The remaining point I speak of is the intrinsic assumption that is used for the entire experiment. Instead of treating general noise, the authors still need to assume that the state has a specific form, that of an isotropic state. It is inherent in the method, as it is exactly the goal of the authors to fit p (the isotropic noise parameter) through minimal experimental effort. This will always limit the method to be an estimation technique, and I don't see how one can practically certify entanglement (or its dimension) using such assumptions. The advantage of witnesses would be that they can directly be used as certificates for adversarial protocols, as they do not make assumptions about the source/channels (i.e. states). That being said, having a quick first estimate, before proceeding to more involved techniques has its merits. And the authors are right that this first estimate compares well against quantum state tomography.

Thank you for this very balanced summary. As you correctly point out, the “minimal experimental effort” for a “quick first estimate” are key. Yes, we demonstrate the approach using the isotropic state. But the idea behind the work, the projectors and the resultant visibilities, are independent of this in the sense that the state never comes into the equation. It is only when extracting information from the visibilities that we ask what we are interested in. In this sense one could replace the state with anything provided it accurately describes the system in question. This is true for virtually all methods.

Further, our main goal was not to certify entanglement but rather to estimate the number of modes that constitute the dimensions of the pure part of the state. Accordingly, our approach can determine the size of the Hilbert space that a quantum system can occupy (capacity) and its purity (quality), and given this information one can make informed

decisions about how to perform purification, entanglement concentration processes, etc. Therefore, it has practical significance.

As for the certification, we hoped that showing the link between the Fidelity witness and our approach could convince the reviewer that there is useful information that one could extract from our measurements. As is, our technique reveals the benefit of using simple conditional measurements (visibilities), therefore making it easy to implement. The visibilities we measure probe the state and as shown in the simulations scale with dimensionality, but the assumptions are only made in the final analyses while using them to infer information about the state that produces them. Just as with maximum likelihood estimation in quantum state tomography, the model of the state is essential in accurate prediction.

We believe that our approach adds to the toolbox of high-dimensional quantum state characterisation rather than diminishing the value of other measures.

But what about direct measurements of fidelity (scales as d) or even just two unbiased measurements? Is it really faster?

Yes, it really is. Our approach indeed accurately characterises a quantum state, and like any other technique has its limitations. The true benefit here is the shortened measurement time, which is far less than the 2 MuB approach as it scales linearly ($d/2$) instead of quadratically with dimensions (we have used both in our lab so we know!). The method requires only $2d$ types of measurements, they actually mean to say that it requires $2d^2$ projections that form 2 mutually unbiased observables. We hope you will find the toned down revision an acceptable account of what is possible and what our approach offers.

It is as you said: one would like to use our tool to quickly gauge the situation and extract important parameters, and then do the reduced QST with 2 MUBs to probe deeper: horses for courses.

As a side-remark, I think the authors did well in also estimating the Schmidt number. I don't agree that, as the authors write '**K reveals the entangled dimensions ...**'. K reveals the inverse purity. It happens to be 1 for separable states and d for states entangled in all dimensions, but in-between there is no operational connection between the statement from the authors and the meaning of that number. There are states where all dimensions certifyably contribute to the entanglement, yet K is close to 1. Due to the monotonicity of Renyi entropies, I believe that K is a lower bound to the Schmidt rank and thus dimensionality, but it can't be reasonably equated to 'entangled dimensions' more than any other entropy (e.g. why $\text{Tr}(\rho^2)$? Why not $\text{Tr}(\rho^3)$? They all have a similar merit in quantifying entanglement).

We completely agree with the reviewer regarding this point. What you picked up was an instance that we missed. We have corrected this throughout the manuscript now.

My conclusion is that is a great experiment, good and honest science, but my prediction is

that the method makes too many assumptions to likely be adopted. But I would also be happy to be proven wrong and ultimately only time will tell.

We really appreciate the thoughtful comments you have made and the constructive and balanced nature of them too. We believe that this revised version makes clear the advantages of our approach, tones down the criticism of other methods, and focusses on the value it can bring to the toolbox. As experimentalists we have used most techniques in the literature and we find the proposed approach here useful because it is so fast and easy to implement. The idea that you can “quickly” test a 100 dimensional state is surely going to attract attention and inspire future studies and advances. We hope to have provided enough evidence to sway you that it is worth sharing with the community of readers of this journal.

Reviewer #2 (Remarks to the Author):

I am impressed by the diligent work done by the authors. I feel that all of my concerns have been well addressed in the response Letter and in the revision. Thus I would like to recommend its publication in Nature Communications.

We are pleased that reviewer finds our work exceptional enough for Nature Communications.

Reviewer #3 (Remarks to the Author):

This manuscript has certainly improved after a revision, especially new data reporting on a measurement in a different type of spatial basis (coordinate vs OAM). I now could imagine the concept and the measurement idea, which I find interesting and elegant, to be of interest for the broad reader audience of Nature Communications, although I doubt it will be a widely adopted method of quantum state characterization.

We thank the reviewer for the positive assessment that this work would be of interest to the broad readership of this journal. In our general response we have tried to motivate why we find it useful in our own lab, and why we believe it will be adopted by others, perhaps as part of a composite toolkit.

The purities reported in the new results are again uncharacteristically low when compared to similar measurements, including a number of those that do not subtract background noise. Comparing, for example, with the pixel-entanglement work mentioned in the response [Quantum 4, 376 (2020)], where state fidelities of 95% are reported for dimensions of up to 20, while the new data of the manuscript reports purities (and fidelities via Eq.40) that are essentially almost zero for anything with a dimension above 9. Thus, the demonstration does show qualitative dependence but fails to perform as a quantitative measurement.

Although the purities that we report are low, we believe that they are accurate. We deliberately make use of high noise levels, so that this appears in the data. Our goal was to demonstrate the technique – not report the highest number of dimensions of entanglement.

In the paper, [Quantum 4, 376 (2020)], the goal was to report the very highest level of entanglement, so the results were taken under pristine conditions. One significant difference between that work and ours is the gate time of the coincidence detection. Our gate time was 25 ns, so it is no surprise that our measured and reported states are less pure than one with a shorter gate.

It is for this reason that we performed an additional experiment with a shorter gate for the purpose of this response, see figure below. In this new experiment, we used our approach on another of our quantum experiments, with a lower gating times (1.5 ns) and higher integration times (30 s) resulting in purities of up to 80-90% (see example below). The higher purities here are a result of the different equipment, and not that we did not measure the state accurately in the first place. This gives us confidence that our results are true, and they correctly reflect information about the measured state.

I would like to reiterate that part of the problem, at least in the OAM case, lies in the fact that this measured state cannot be modelled accurately as a Werner-like state with isotropic noise. The amount of observed/measured noise is higher for the parts of the state formed by larger OAM values. This is a well-known fact, demonstrated multiple times since a long time ago including by some of the co-authors of this manuscript [New J. Phys. 11 103024 (2009)].

We agree with the reviewer regarding this point, in-fact one sees this from our spectrum plots in Figure 3. While this shows that perhaps the model needs to be modified to take this into consideration, the higher OAM crosstalk still isn't the main contribution especially as the environmental noise increases. Since the measurement approach (how you build the projectors and measure) is decoupled from the state, it should be possible to add crosstalk and mode dependent noise into the model and would require a more detailed study which we will consider in future. We have added text to reflect that the parameters that the minimisation is performed over could be changed to incorporate this in the model.

Finally, given the recent advances in adaptive and self-guided tomography [PRL 117, 040402 (2016); PRA 101, 022317 (2020); PRL 126, 100402 (2021)], the protocol of Bavaresco et al. for dimensionality witnessing, and the textbook SWAP test [Opt. Express 26, 8443 (2018); arXiv:2103.10219], for direct measurement of the purity of any quantum state, I find the scaling comparison with the standard QST to be forced.

Thank you for bringing this to our attention – we now cite the relevant papers we missed (we did not cite the qubit PRL paper). Using purity as an example, you can extract it from careful QST (effort in the detection) and/or by direct measure after making two copies of the system (effort in the preparation). We are not negating these; instead we are offering a quick and easy test to determine what it is before making decisions on more thorough approaches. We have now revised the discussion section in the context of the prior work and highlighted the speed of our approach but in a more toned down fashion. We want to recognise the wealth of excellent research on quantum state measurements, and we want to make it clear what the pros and the cons of our method are. We hope that the reviewer recognises that we are not trying to force any scaling comparison now – we are trying to be objective with regards to the benefits.

Reviewers' Comments:

Reviewer #1:

Remarks to the Author:

I want to thank the authors again for their final amendments. I believe the article is now fully correct and the communication transparent. Whether the method will work beyond very simple states (such as the exemplary isotropic state used) remains to be seen, but I think the article is in a form that could be published and time will tell whether the speed-up in estimating fidelity will yield practical advantages for characterising high-dimensional states.

Reviewer #3:

Remarks to the Author:

I would be happy now to see this work published Nature Communications. A very large coincidence window (this information was in the manuscript since the beginning, but I haven't previously noticed it) is indeed a valid explanation for an increased level of noise in a CW pumped SPDC experiment. The new clarifications about the state model and the method applicability, now added to the concept and discussion sections made this work more rigorous and precise. With the two types of measurement, OAM and pixel-based, the appeal of this method has been well demonstrated.

REVIEWER COMMENTS AND RESPONSES

Reviewer #1:

I want to thank the authors again for their final amendments. I believe the article is now fully correct and the communication transparent. Whether the method will work beyond very simple states (such as the exemplary isotropic state used) remains to be seen, but I think the article is in a form that could be published and time will tell whether the speed-up in estimating fidelity will yield practical advantages for characterising high-dimensional states.

We thank the reviewer for recommending our work for publication. We also owe our gratitude to the referee since their inputs helped us improve the quality and transparency of our manuscript. We hope that readers will be attracted to our technique mainly for its practicality and advancing measurement techniques for high dimensional quantum states.

Reviewer #3 :

I would be happy now to see this work published Nature Communications. A very large coincidence window (this information was in the manuscript since the beginning, but I haven't previously noticed it) is indeed a valid explanation for an increased level of noise in a CW pumped SPDC experiment. The new clarifications about the state model and the method applicability, now added to the concept and discussion sections made this work more rigorous and precise. With the two types of measurement, OAM and pixel-based, the appeal of this method has been well demonstrated.

We thank the reviewer for finally recommending our work for publication. The reviewers' concerns have truly helped us to improve our manuscript and we appreciate their inputs and time taken to review this work.